# Epidemiology of yellow fever virus in humans, arthropods, and non-human primates in sub-Saharan Africa: A systematic review and meta-analysis

Martin Gael Oyono[1,2], Sebastien Kenmoe[3]*, Ngu Njei Abanda[4], Guy Roussel Takuissu[5], Jean Thierry Ebogo-Belobo[6], Raoul Kenfack-Momo[7], Cyprien Kengne-Nde[8], Donatien Serge Mbaga[9], Serges Tchatchouang[10], Josiane Kenfack-Zanguim[7], Robertine Lontuo Fogang[11], Elisabeth Zeuko'o Menkem[12], Juliette Laure Ndzie Ondigui[9], Ginette Irma Kame-Ngasse[6], Jeannette Nina Magoudjou-Pekam[7], Arnol Bowo-Ngandji[9], Seraphine Nkie Esemu[3], Lucy Ndip[3]

1 Centre for Research on Health and Priority Pathologies, Institute of Medical Research and Medicinal Plants Studies, Yaounde, Cameroon, 2 Laboratory of Parasitology and Ecology, Department of Animal Biology and Physiology, University of Yaounde I, Yaounde, Cameroon, 3 Department of Microbiology and Parasitology, University of Buea, Buea, Cameroon, 4 Virology Department, Centre Pasteur of Cameroon, Yaounde, Cameroon, 5 Centre for Food, Food Security and Nutrition Research, Institute of Medical Research and Medicinal Plants Studies, Yaounde, Cameroon, 6 Medical Research Centre, Institute of Medical Research and Medicinal Plants Studies, Yaounde, Cameroon, 7 Department of Biochemistry, The University of Yaounde I, Yaounde, Cameroon, 8 Epidemiological Surveillance, Evaluation and Research Unit, National AIDS Control Committee, Douala, Cameroon, 9 Department of Microbiology, The University of Yaounde I, Yaounde, Cameroon, 10 Scientific Direction, Centre Pasteur of Cameroon, Yaounde, Cameroon, 11 Department of Animal Biology, University of Dschang, Dschang, Cameroon, 12 Department of Biomedical Sciences, University of Buea, Buea, Cameroon

* sebastien.kenmoe@ubuea.cm

**Data Availability Statement:** All relevant data are within the manuscript and its Supporting information files.

## Abstract

Yellow fever (YF) has re-emerged in the last two decades causing several outbreaks in endemic countries and spreading to new receptive regions. This changing epidemiology of YF creates new challenges for global public health efforts. Yellow fever is caused by the yellow fever virus (YFV) that circulates between humans, the mosquito vector, and non-human primates (NHP). In this systematic review and meta-analysis, we review and analyse data on the case fatality rate (CFR) and prevalence of YFV in humans, and on the prevalence of YFV in arthropods, and NHP in sub-Saharan Africa (SSA). We performed a comprehensive literature search in PubMed, Web of Science, African Journal Online, and African Index Medicus databases. We included studies reporting data on the CFR and/or prevalence of YFV. Extracted data was verified and analysed using the random effect meta-analysis. We conducted subgroup, sensitivity analysis, and publication bias analyses using the random effect meta-analysis while $I^2$ statistic was employed to determine heterogeneity. This review was registered with PROSPERO under the identification CRD42021242444. The final meta-analysis included 55 studies. The overall case fatality rate due to YFV was 31.1% (18.3–45.4) in humans and pooled prevalence of YFV infection was 9.4% (6.9–12.2) in humans. Only five studies in West and East Africa detected the YFV in mosquito species of the genus *Aedes and in Anopheles funestus*. In NHP, YFV antibodies were found only in

**Funding:** This project is part of the EDCTP2 programme supported by the European Union under grant agreement TMA2019PF-2705 to SK (http://www.edctp.org/projects-2/edctp2-projects/edctp-preparatory-fellowships-2019/#). The funders had no role in study design, data collection and analysis, decision to publish, or preparation of the manuscript.

**Competing interests:** The authors have declared that no competing interests exist.

members of the *Cercopithecidae* family. Our analysis provides evidence on the ongoing circulation of the YFV in humans, *Aedes* mosquitoes and NHP in SSA. These observations highlight the ongoing transmission of the YFV and its potential to cause large outbreaks in SSA. As such, strategies such as those proposed by the WHO's Eliminate Yellow Fever Epidemics (EYE) initiative are urgently needed to control and prevent yellow fever outbreaks in SSA.

## Author summary

Yellow fever, one of the most feared lethal zoonotic disease is re-emerging as a public health threat to tropical and sub-tropical countries of South America and Africa. Despite the existence of an effective yellow fever vaccine that is administered through mass vaccination campaigns and in routine immunization programs. against this disease, the mortality remains very high during the outbreak of yellow fever in several sub-Saharan African (SSA) countries. It is necessary to have accurate epidemiological data on YFV infection, in order to prioritize the policies, funding for public health interventions, and health-care planning. Our study is the first systematic review and meta-analysis with data provided on the case fatality rate (CFR) and prevalence of YFV in humans, and prevalence of YFV in arthropods, non-human primates (NHP), and other animal species in SSA. Broadly, the study shows that the CFR and prevalence of YFV in humans were relatively high and low respectively. Furthermore, mosquitoes of the genus *Aedes* and *Anopheles funestus* were the main vectors of YFV. Finally, only NHP of the *Cercopithecidae* family were the reservoirs of the YFV in SSA. These data provide evidence on the ongoing circulation of the YFV in SSA and the possibility of the large outbreaks YFV in SSA. Author suggests that preventive strategies should be promoted by educating and raising people's awareness about YFV infection and strengthening practitioners 'capacities towards adequate diagnosis and proper management of this infection in SSA.

## Introduction

Today and throughout history, animals have been an important source of pathogens transmitted to humans [1]. Between 1940 and 2004, over 60% of emerging infectious diseases in humans were due to pathogens from wildlife or domestic animals [2]. Some animal pathogens must be transmitted through an insect vector in order to infect humans [3,4]. A good example of such an animal pathogen requiring an insect vector to infect humans is the yellow fever virus (YFV) that causes yellow fever (YF) in humans. The YFV is an enveloped, positive-sense, single-stranded RNA virus that belongs to the genus of *Flavivirus* of the family Flaviviridae [5].

In a majority of humans, infection with the YFV may be asymptomatic or present with mild fever, headache, muscle pain and nausea. However, in about 20% of infected humans, the infection will progress to a severe form characterized by high fever, bleeding, jaundice, shock and, multiorgan failure and about 50% of these severe patients die within 7 to 10 days [6,7]. According to the World Health Organization (WHO), each year, about 200,000 YF cases and 30,000 deaths are reported of which nearly 90% of cases and death occur in Africa [8]. Clinically, YF presents with signs and symptoms similar to those of other diseases such as viral hepatitis and malaria and could easily be misdiagnosed for these diseases. As such, the actual number of YF cases could be several times higher than the reported number of cases [9].

YF is endemic in tropical and sub-tropical countries of South America and Africa [10]. In Africa, the YFV is spread through three transmission cycles: the sylvatic (or jungle) cycle, the intermediate (savannah) cycle and the urban cycle [11]. The sylvatic cycle is the primary cycle and spread of the YFV occurs between non-human primates (NHP) as reservoir hosts and forest dwelling mosquitoes such as *Aedes africanus*, *Aedes opok*, *Aedes simpsoni*, *Aedes luteocephalus*, *Aedes taylori*, *Aedes vittatus*. The YFV can then be passed from NHPs to humans when they visit or work in the jungle. The savannah cycle involves transmission of the YFV by an anthropo-zoophilic mosquito such as *Aedes albopictus* to humans living or working at the fringes of the jungle area. *Aedes albopictus* feeds on both animals and humans and serves as a bridge vector transferring YFV from animals to humans. In the urban cycle, domestic mosquitoes such as *Aedes aegypti* facilitates human-to-human transmission of the YFV [11–13].

The existence of an effective YF vaccine since the 1930s has greatly helped to contain YF outbreaks in Africa and beyond. However, since the mid-2000s, an upsurge in YFV transmission events have been reported throughout YF endemic countries especially in Africa [14–16]. Giving this unusual resurgence of YF transmission and the likelihood of major outbreaks, information on the burden and prevalence of YF in Africa is necessary for the development and deployment of counter measures [15]. We conducted a systematic review and meta-analysis to provide information in sub-Saharan Africa (SSA) on the prevalence and case fatality rate of YF in humans and on the prevalence of YF in arthropods and NHP.

## Methods

### Protocol and inclusion criteria

This systematic review was designed and conducted using the Preferred Reporting Items for Systematic Reviews and Meta–analyses (PRISMA) checklist (S1 Table) [17]. The protocol of this systematic review was published in the international database PROSPERO under the identification CRD42021242444. As this review reports on previously published data, ethical clearance was not required.

In this review, we considered cross-sectional studies, community and hospital-based studies and outbreak reports carried out in SSA. We included studies that reported on the prevalence and case fatality rate of YFV in humans, arthropods, NHP, and other animal species in SSA. Humans were further classified according to age, gender, disease state (healthy individuals and suspected YFV cases) and grouped according to the inclusion criteria of the selected studies. Arthropods were mosquitoes of the Diptera order. Other animal species and NHP were classified into their specific taxonomic orders.

To determine YF prevalence within the human populations, we included all studies reporting i) the detection of YFV specific IgM, ii) the detection of YFV specific IgG and/or IgM, iii) detection of YFV specific neutralizing antibodies, iv) detection of YFV by RT-PCR or viral isolation, v) detection of YFV specific antigens. As such, we included studies that utilize a wide range of methods for detection such as indirect immunofluorescence, complement fixation, culture, RT-PCR or ELISA, next-generation sequencing, or western blot.

Studies carried out of SSA or with inappropriate study design (comments, case reports, reviews, systematic reviews, and meta-analyses) as well as studies using positive YF samples or with sample size ≤ 10 were excluded. All studies from which data on YFV prevalence and case fatality rate (CFR) could not be obtained were also excluded. Studies published in neither the English nor French language or for which full text and abstract were either not available or could not be retrieved were also excluded.

### Article search method to identify studies for inclusion

A comprehensive strategy was designed to enable a search of relevant studies in the following databases: PubMed, Web of Science, African Journal Online, and African Index Medicus. These databases were searched for studies published in English or French languages from January 2000 until February 2021 and updated in March 2022 to have contemporary data on YF. Main search strategy (S2 Table) was developed and used to search the databases. The literature search was supplemented by a review of the reference list of identified articles to find additional potential studies. Names of SSA countries and regional groupings were also used to search for studies indexed under these names.

### Study selection and data extraction

After removing duplicates from the list of studies, titles and abstracts of the eligible studies were independently examined by two study authors (JETB and SK) for the selection of relevant studies. Data from the included studies was extracted using the online google form by 10 study authors and verified by SK and MGO. The extracted data were: the name of the first author, the year of publication, the study design, country, study period, sampling method, the study population (ill or apparently healthy humans, individual mosquitoes or negative pools, and NHP), age range of study population, YFV vaccine status, WHO Region, UNSD Region, country income level, YFV detection assay, YFV marker detected (virus, antigen, RNA, IgM or IgG), type of sample used for YFV detection, infection status (current, recent or past infections), sample size, number of positive for YFV, and number of deaths within YFV positive. For studies with less than 10 participants in animals and mosquitoes tested in pools and reporting a positive result, the names of the positive species were retained. Any difference in opinion with regards to the selection and inclusion of studies and extracted data were resolved by discussion, consensus, or by a third author.

### Evaluation of the quality of studies

The quality of the studies considered was assessed using the tool of Hoy et al. (S3 Table) [18]. This tool consists of 10 questions that evaluates the external and internal validity of the study. The expected answers for each question were "yes", "no", "unclear" and "not applicable" depending on the content of the articles. A score of 1 was assigned for all "yes" answers and 0 for the other ones. Articles with a total score of 0–3, 4–6, and 7–10 were considered to be respectively at high, moderate, and low risk of bias.

### Data synthesis

Data analysis was performed using R version 4.1.0 software [19]. We described without meta-analysis the positive YFV species of mosquitoes, NHP, and other animal species. The random effect model was performed to estimate combined prevalence of YFV and/or CFR in humans [20]. The Freeman-Tukey Double arcsine transformation was performed for the prevalence calculation [21]. The prevalence were represented as a forest plot with their corresponding 95% confidence intervals (CI). For the plot of the forest plot, a weighting according to the size of the sample was carried out to determine the size of the diamonds [22]. The Clopper-Pearson method was performed to calculate the 95% confidence interval of the prevalences [23,24]. Prevalences for future studies were determined by calculating a prediction interval [22,25]. The Cochran Q test and the $I^2$ test statistic were used to measure the magnitude of heterogeneity between the included studies. The value of $I^2$ more than 50%, was an indication of significant heterogeneity in the studies [26,27]. Sources of heterogeneity were investigated by

subgroup analysis, and the sensitivity analysis that included only cross-sectional and low risk of bias studies was performed. Visual inspection of a funnel plot and the Egger test were used to estimate the risk of publication bias [28].

## Results

### Study selection

The literature search through databases provided a total of 3888 potentially relevant articles. After removing 1287 duplicate articles and excluding 2389 articles based on a careful review of the titles and abstracts, the remaining 212 articles were assessed for eligibility. Of the 212 articles, 157 full text articles were excluded for multiple reasons with absence of data on YFV prevalence or case fatality rate being the predominant reason (Fig 1 and S4 Table). We include a final total of 55 articles (151 datapoint on prevalence and/or CFR) in the qualitative and quantitative synthesis for this review [3–9,14,29–75].

### Assessment of study quality

The majority of studies included were at moderate risk of bias, 119/151 (78.8%); a few had a low risk of bias 32/151 (21.2%), and none of the included studies had a high risk of bias (S5 Table).

### Baseline characteristics of included studies

The summary and individual data of the included studies are presented in S6 and S7 Tables. The studies were published between 2001 and 2022 and the participants were recruited between 1990 and 2021. From the 151 data reported, 4/151 (2.8%) data reported CFR of YFV in humans, 71/151 (45.1%) reported prevalence of YFV in humans, 65/151 (43.0%) in mosquitoes, 7/151 (4.9%) in NHP, and 4/151 (2.8%) in other animal species. Data were majority recorded on cross-sectional studies 135/151 (88.7%), non-probabilistic 137/151 (90.7%), prospective 134/151 (88.7%), multicenter 136/151 (90.1%), and community-based 115/151 (76.2%). The included studies were conducted in 18 SSA countries with the highest number of studies conducted in Senegal 43/151 (28.5%) and the Central African Republic 33/151 (21.9%). Countries in West Africa 63/151 (41.7%) and Central Africa (49/151; 32.5%) had the largest number of studies. The predominant detection assays used to detect YFV in the included studies were sandwich ELISA 38/151 (25.2%), real-time RT-PCR 37/151 (24.5%), and infection of cellular cultures 35/151 (23.2%). The majority of studies found current YFV infection evidenced by the detection of viral RNA or live virus 89/151 (58.9%) and recent infection evidenced by the detection of IgM antibodies 32/151 (21.2%). With respect to studies among humans 64/63 (98.4%), NHP 5/7 (71.4%) and other animals 4/4 (100%), most of them found YFV in serum sample.

### Prevalence of yellow fever virus in mosquitoes, non-human primates, and other animal species in sub-Saharan Africa

The mosquitoes tested in the pool were from the Culicidae family. Individually tested mosquitoes from the genus *Aedes* 28/65 (43.1%), *Culex* 14/65 (21.5%), and *Anopheles* 13/65 (20.0%) were predominantly represented. Only five articles reported the detection of YFV in individually tested mosquitoes with prevalence ranging from 0.0 to 12.0%. The mosquito species that were positive for YFV included *Aedes aegypti*, *Aedes africanus*, *Aedes centropunctatus*, *Aedes dalzieli*, *Aedes furcifer*, *Aedes Luteocephalus*, *Aedes mcintoshi*, *Aedes taylori*, *Aedes vittatus*, and *Anopheles funestus* [29,39,41,42,46]. All studies involving mosquitoes were detection of the

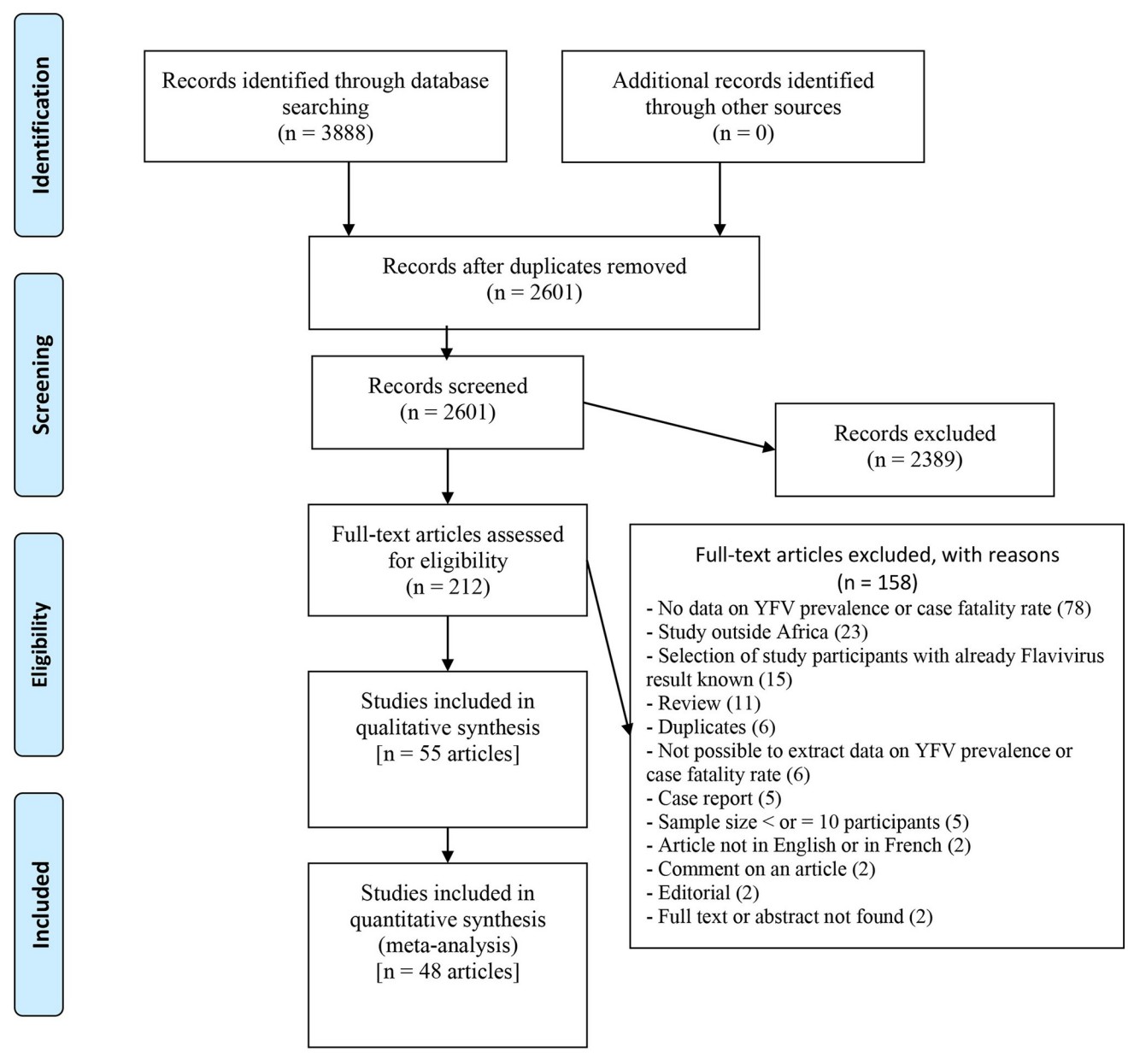

**Fig 1. PRISMA flow diagram.**

ongoing presence of YFV either by virus isolation or RT-PCR. Mosquito samples positive for YFV were only reported in Senegal and Kenya. Only two articles reported the detection of YFV antibodies in NHP with prevalence ranging from 1.8 and 48.0%. The family of *Cercopithecidae* 6/7 (85.7%) was the most represented NHP. YFV antibodies were detected in Gabon and in the Central African Republic in *Cercopithecidae* of the species Mandrills, *Chlorocebus*, *Cercopitheque*, *Cynocephalus*, and *Erythrocebus* [4,70]. None of the studies detected the YFV RNA (current infection) in NHP tested. Other animal species studied to date belong to the order of *Artiodactyla* 2/4 (50%), *Chiroptera* 1/4 (25.0%) and Proboscidea 1/4 (25.0%). Only one sample from bats in eastern Africa (Uganda) was found to be positive for YFV antibodies

[51]. No sample from other animal species (Buffalo, Duike and Elephant) from countries in Central Africa region were positive for YFV or YF antibodies.

## Results of the meta-analysis

### Case Fatality Rate of Yellow Fever Virus infection in humans in sub-Saharan Africa

A case fatality rate of YFV infection was recorded in 4 studies conducted in four Africans countries: Democratic Republic of the Congo [76], Nigeria [62], Uganda [5] and Sudan [6] (Fig 2a and 2b). A total of 128 YFV suspected cases were recruited in the 4 studies giving an overall CFR of 31.1% (95% IC: 18.3–45.4) and data presented no heterogeneity ($I^2 = 48.6\%$, [95% CI = 0.0%–83.0%], $P = 0.1197$) (Fig 3). This estimated CFR of YFV varied across infection status with 29.8% [95% CI = 12.7–49.9] in people with current infection, and 37.0% [95% CI = 19.6–56.6] in people with recent infection. Based on the funnel plot (S1 Fig) and Egger's regression test, there was a good symmetry and no evidence of potential publication bias ($P = 0.382$) for determining the CFR of YFV in humans.

### Prevalence of Yellow Fever Virus infection in humans in sub-Saharan Africa

Studies on humans recruited mostly YFV suspected cases 22/71 (34.4%), apparently healthy individuals 22/71 (31.0%) and febrile patients 14/71 (19.7%). None of the studies considered, reported on the vaccination status of the enrolled participants. The overall prevalence of YFV recorded in 67098 human participants recruited from 71 datapoints prevalence was 9.4% (95% CI = 6.9–12.2) with a substantial heterogeneity between studies ($I^2 = 99.1\%$ [95% CI = 99.0%–99.2%], p <0.001) (Figs 2c, 2d, 2e and 4, and S2 Fig). Regardless of the type of

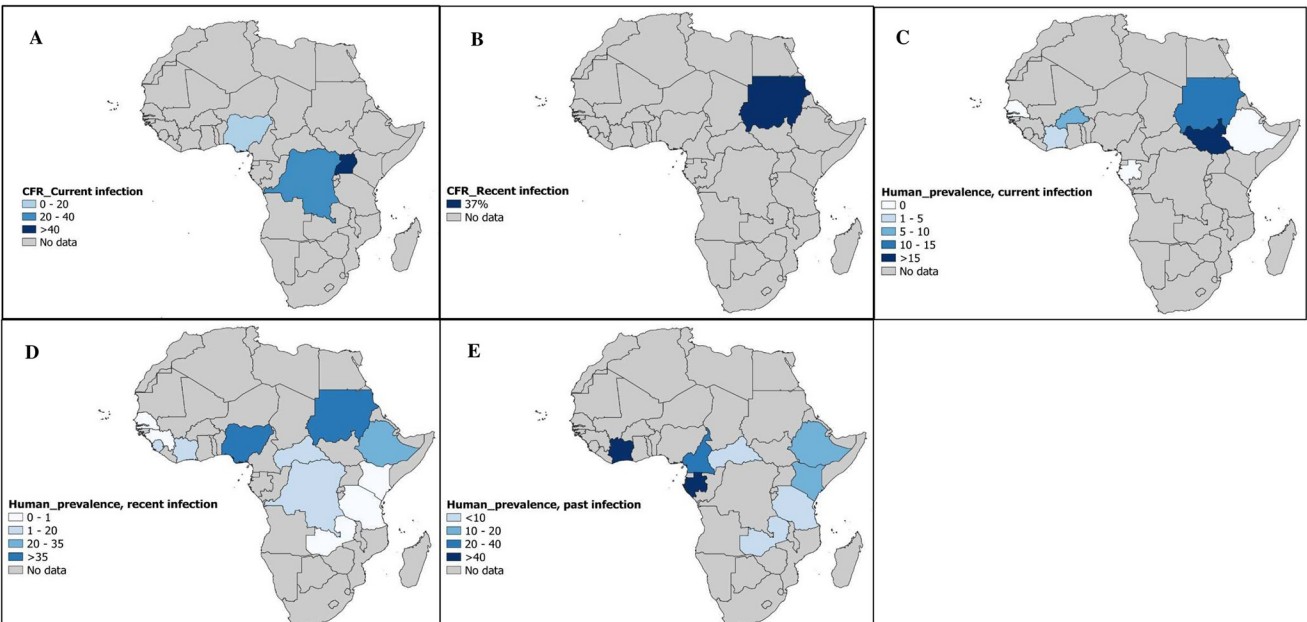

**Fig 2. Case fatality rate and prevalence estimate of yellow fever virus in humans in sub-Saharan Africa.** The letters (A and B) show the case fatality rate in humans with current and recent yellow fever virus exposures, respectively. The letters (C, D, and E) denote current, recent and past yellow fever virus exposures, respectively. The base map was taken from (https://www.naturalearthdata.com/) and modified with Qgis software.

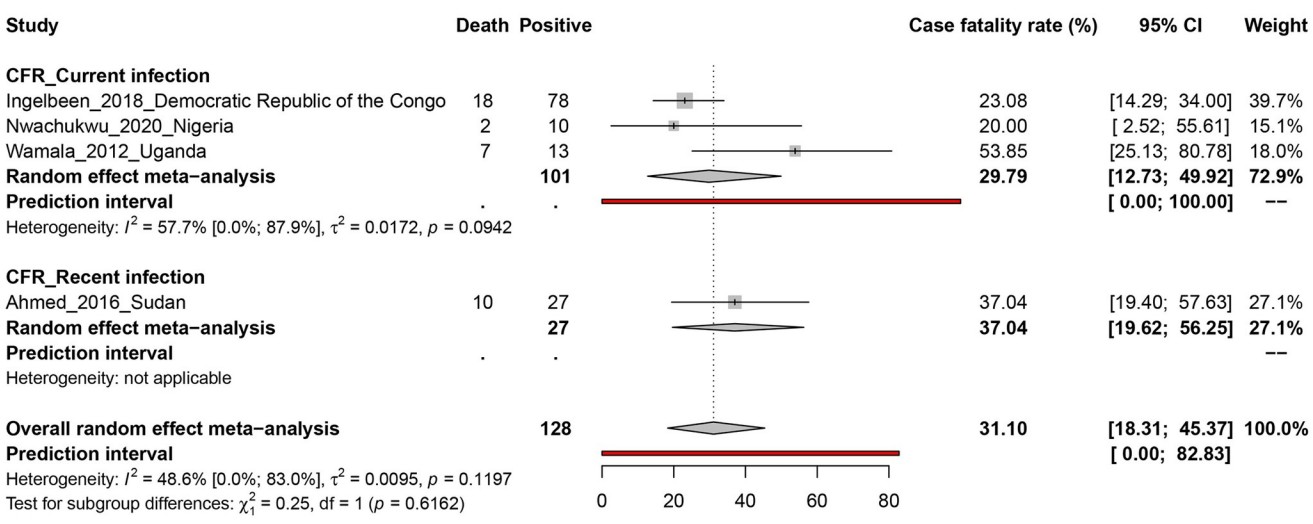

**Fig 3. Case fatality rate estimate of yellow fever virus infections in humans in sub-Saharan Africa.**

infection, the prevalence of YFV was 18.8% (95% CI = 11.8–27.0; 15578 participants), 6.0% (95% CI = 3.4–9.2; 29267 participants), and 5.3% (95% CI = 2.7–8.5; 22053 participants) in human participants with past, recent, and current infection respectively. Funnel plot (S3 Fig) and Egger's regression test (Table 1) showed the existence of publication bias for studies of all types of YFV infections (p<0.001).

## Subgroup analysis of meta-analysis results for case fatality rate and prevalence of yellow fever virus in humans in sub-Saharan Africa

Subgroup analyses of case fatality rate and prevalence of YFV in humans, mosquitoes, NHP, and other animal species in SSA is summarize in S8 Table and Fig 2. Analysis of the data

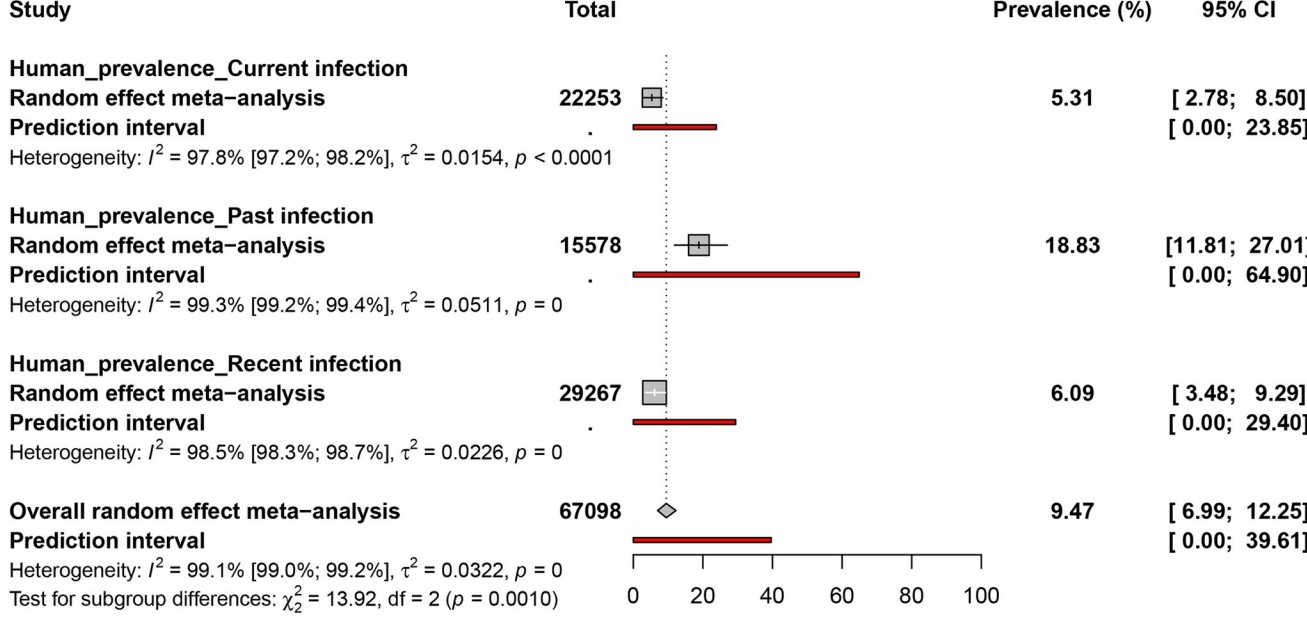

**Fig 4. Prevalence estimates of yellow fever virus infections in humans in sub-Saharan Africa.**

**Table 1. Summary of meta-analysis results for case fatality rate and prevalence of yellow fever virus in humans in sub-Saharan Africa.**

| | Prevalence. % (95%CI) | 95% Prediction interval | N Studies | N Participants | ¶H (95%CI) | §I² (95%CI) | P heterogeneity |
|---|---|---|---|---|---|---|---|
| YFV case fatality rate in humans | | | | | | | |
| Current infection | | | | | | | |
| Overall | 29.8 [12.7–49.9] | [0–100] | 3 | 101 | 1.5 [1–2.9] | 57.7 [0–87.9] | 0.094 |
| Cross-sectional | 23.1 [14.3–33.1] | NA | 1 | 78 | NA | NA | 1 |
| Recent infection | | | | | | | |
| Overall | 37 [19.6–56.2] | NA | 1 | 27 | NA | NA | 1 |
| YFV prevalence in humans | | | | | | | |
| Current infection | | | | | | | |
| Overall | 5.3 [2.8–8.5] | [0–23.8] | 19 | 22253 | 6.7 [6–7.4] | 97.8 [97.2–98.2] | <0.001 |
| Cross-sectional | 3.6 [1.3–6.8] | [0–20.8] | 12 | 21313 | 7.7 [6.8–8.8] | 98.3 [97.8–98.7] | <0.002 |
| Low risk of bias | 0.8 [0–3.1] | [0–19.2] | 4 | 18272 | 6.8 [5.3–8.9] | 97.9 [96.4–98.7] | <0.003 |
| Past infection | | | | | | | |
| Overall | 18.8 [11.8–27] | [0–64.9] | 22 | 15578 | 11.7 [10.9–12.6] | 99.3 [99.2–99.4] | <0.001 |
| Cross-sectional | 18 [11.2–26] | [0–62.4] | 21 | 14973 | 11.3 [10.5–12.2] | 99.2 [99.1–99.3] | <0.002 |
| Low risk of bias | 12.7 [5–23.3] | [0–65.7] | 13 | 10297 | 13.5 [12.4–14.6] | 99.4 [99.3–99.5] | <0.003 |
| Recent infection | | | | | | | |
| Overall | 6.1 [3.5–9.3] | [0–29.4] | 30 | 29267 | 8.3 [7.7–8.9] | 98.5 [98.3–98.7] | <0.001 |
| Cross-sectional | 4.3 [2.1–7.2] | [0–24.1] | 25 | 28353 | 8.2 [7.5–8.9] | 98.5 [98.2–98.7] | <0.002 |
| Low risk of bias | 2.1 [0.9–3.9] | [0–11.1] | 11 | 24532 | 6.4 [5.5–7.5] | 97.6 [96.7–98.2] | <0.003 |

CI: confidence interval; N: Number; 95% CI: 95% Confidence Interval; NA: not applicable.

¶H is a measure of the extent of heterogeneity. a value of H = 1 indicates homogeneity of effects and a value of H >1indicates a potential heterogeneity of effects.

§: I2 describes the proportion of total variation in study estimates that is due to heterogeneity. a value > 50% indicates presence of heterogeneity

showed that YFV CFR was significantly higher in community-based studies (42.3%; 95%CI: 27.0–58.4; p = 0.031). Community-based studies (10.4%; 95%CI: 6.9–14.4; p = 0.015), conducted in Nigeria (44.7%; 95%CI: 36.3–53.2), Cameroon (26.1%; 95%CI: 12.0–43.1), and Sudan (25.6%; 95%CI: 5.8–52.5; p < 0.015), recruiting hospitalized participants (27.1%; 95% CI: 11.3–46.5; p = 0.014), suspected YFV cases (11.4%; 95%CI: 7.7–15.7), YFV positive case contacts (10.6%; 95%CI: 0.0–34.2), and apparently healthy individuals (10.8%; 95%CI: 6.1–16.7; p < 0.001) were more likely to show higher YFV prevalence.

## Discussion

The present study is the first systematic review and meta-analysis on the prevalence and CFR of YFV in humans, and YFV prevalence in arthropods, NHP, and other animal species in SSA. Overall, our analysis reports a computed pooled CFR estimate due to YF of 31.1% in humans and an overall prevalence of 9.4% of YFV in humans. Mosquitoes positive for YFV included several species of the genus *Aedes* and *Anopheles funestus*. Only NHP of the Cercopithecidae family showed serological evidence of exposure to YFV.

The estimated CFR of YFV in humans identified in this review is consistent with that recently reported in a global review with a CFR of 36% for Africa [77]. Such a high CFR could be due to delays or deterrents in seeking care during the early less sever phase of the disease or delayed clinical diagnosis of cases [62,76]. It should also be noted that African population is more at risk of contracting the yellow fever virus and of developing severe forms and death due to a low rate of vaccination coverage and daily activities that bring them closer to vectors such as agriculture, livestock, hunting, and deforestation [77–80]. Also, the existence in Africa of other health conditions such as malnutrition, tuberculosis, malaria and, HIV are other factors that could be associated with this high of YFV CFR [81–83]. About 1/5th (18.8%) of sampled human participants included in this review had IgG antibodies against YFV (past infection). YF IgG antibodies could be naturally acquired following an infection with the virus or following vaccination with the YF vaccine [84]. It is unclear if participants in the included studies had received the YF vaccine as participant's vaccination history was not reported in most studies. However, the estimated seroprevalence level is slightly reflective of naturally acquire IgG antibodies as the value is comparable to values reported by individual studies conducted on non-vaccinated persons in subgroup analysis. Even so, we cannot rule out the contribution of the YF vaccine on the seroprevalence levels. Most of the included studies were conducted in countries with moderate to high levels of YF vaccine coverage [85]. Despite reports of a good vaccine coverage in SSA including countries incorporated in this review, YF infection continues to persist. The prevalence rates of current and recent infection were 5.3% and 6.0% respectively. This prevalence levels could even be higher if not of the inherent limitation in detecting YF viral RNA and/or antigen (current infection) and YF IgM antibodies (recent infection). Identifying these infection biomarkers is totally dependent on the timing of sample collection and if the studied area is endemic to other flaviviruses such as Dengue, Zika [86]. YF viral RNA and/or antigen can be detected in serum of symptomatic patients only during the first 7 days of illness or for longer periods in severe cases (30% of patients). As such, in the 70% of patients presenting mild symptoms, if this period of sample collection is missed, testing for viral RNA and/or antigen may not be clinically useful. Generally, YF diagnosis relies on the detection YF IgM antibodies as IgM antibodies can be detected for up to 3 months following infection. However, in patients with a prior history of infection with other flaviviruses, IgM antibodies may absent or present briefly (<1 month) thereby hampering IgM detection [87–89]. Among the included studies, high numbers of current and recent YF infections were predominantly detected in studies in Nigeria, Cameroon, and Sudan and among hospitalized patients who were most likely exhibiting severe symptoms.

Mosquitoes or the genus *Aedes* are the primary vectors responsible for the transmission of the YFV in all transmission cycles [13]. Surprisingly, despite evidence of YF infection in humans, very few of the included individual studies were able to identify YFV in mosquitoes. Broadly, mosquitoes of the *Aedes* genus specifically *Aedes africanus, Aedes furcifer, Aedes luteocephalus, Aedes taylori, and Aedes vittatus* in rural settings and *Aedes aegypti* in an urban setting were found to be positive for YFV. The relatively low detection of YFV in mosquitoes could be due to the low viral load in mosquitoes making detection by direct isolation or RT-PCR very challenging. Also, collected mosquitoes need to be transported at low temperatures to prevent degradation of viral RNA [90]. New techniques and technological advances such as mosquito traps with inserted FTA nuclei acid preservation cards could help bypass some of these challenges in sampling mosquitoes for YF detection [91,92]. NHP are generally considered as competent reservoir hosts of the YFV and are responsible for maintaining the sylvatic YFV transmission cycle [13]. In this review, we found a low rate of YFV antibodies exposure among NHP. Among the NHP, YFV antibodies were detected in mandrills from Gabon and *Cercopithecidae* from the Central Africa Republic, although the number of studies

on NHP were limited. The scanty evidence of acute and/or recent infection of YFV in NHP makes it difficult to understand their role as reservoir host and in maintaining the sylvatic and intermediate transmission cycle in SSA. Although antibodies to YFV were detected in a bat in Uganda, the role of bats as a reservoir could not be ascertained and this would probably be the result of a cross-reaction with another virus [89].

Overall, this systematic review and meta-analysis provides evidence on the ongoing circulation of the YFV in humans, *Aedes* mosquitoes and NHP in SSA. Our analysis reports on the prevalence of the YFV among the different studied populations. The high number of studies included in this review increases the accuracy of reported estimates. However, there are at least two limitations to our study. First, we observed substantial heterogeneity among the included studies that still existed even when subgroup analyses were done. Secondly, most of the reported pooled estimates had significant publication bias. Despite these limitations, our analyses revealed: the presence of YFV in humans with a relatively high CFR especially during outbreak, one family of NHP (Cercopithecidae) served as a potential reservoir host and *Aedes* species as main vector of YFV in SSA. These observations highlight the ongoing transmission of the YFV and its potential causing large outbreaks in SSA. As such, strategies such as those proposed by the WHO's Eliminate Yellow Fever Epidemics (EYE) initiative are urgently needed to control and prevent YFV outbreaks [93].

## Supporting information

**S1 Table. Preferred reporting items for systematic reviews and meta-analyses checklist.**
(PDF)

**S2 Table. Search strategy in PubMed.**
(PDF)

**S3 Table. Items for risk of bias assessment.**
(PDF)

**S4 Table. Main reasons of exclusion of eligible studies.**
(PDF)

**S5 Table. Risk of bias assessment.**
(PDF)

**S6 Table. Characteristics of included studies.**
(PDF)

**S7 Table. Individual characteristics of included studies.**
(PDF)

**S8 Table. Subgroup analyses of case fatality rate and prevalence of yellow fever virus in humans in sub-Saharan Africa.**
(PDF)

**S1 Fig. Funnel chart for publications of the yellow fever virus case fatality rate in sub-Saharan Africa.**
(PDF)

**S2 Fig. Prevalence estimate of yellow fever virus infections in humans in sub-Saharan Africa.**
(PDF)

**S3 Fig. Funnel chart for publications of the yellow fever virus prevalence in humans in Africa.**
(PDF)

## Author Contributions

**Conceptualization:** Martin Gael Oyono, Sebastien Kenmoe, Lucy Ndip.

**Data curation:** Martin Gael Oyono, Sebastien Kenmoe, Guy Roussel Takuissu, Jean Thierry Ebogo-Belobo, Raoul Kenfack-Momo, Cyprien Kengne-Nde, Donatien Serge Mbaga, Serges Tchatchouang, Josiane Kenfack-Zanguim, Robertine Lontuo Fogang, Elisabeth Zeuko'o Menkem, Juliette Laure Ndzie Ondigui, Ginette Irma Kame-Ngasse, Jeannette Nina Magoudjou-Pekam, Arnol Bowo-Ngandji.

**Formal analysis:** Sebastien Kenmoe, Cyprien Kengne-Nde.

**Funding acquisition:** Sebastien Kenmoe.

**Methodology:** Martin Gael Oyono, Sebastien Kenmoe, Ngu Njei Abanda, Guy Roussel Takuissu, Jean Thierry Ebogo-Belobo, Raoul Kenfack-Momo, Cyprien Kengne-Nde, Donatien Serge Mbaga, Serges Tchatchouang, Josiane Kenfack-Zanguim, Robertine Lontuo Fogang, Elisabeth Zeuko'o Menkem, Juliette Laure Ndzie Ondigui, Ginette Irma Kame-Ngasse, Jeannette Nina Magoudjou-Pekam, Arnol Bowo-Ngandji, Seraphine Nkie Esemu, Lucy Ndip.

**Project administration:** Sebastien Kenmoe, Lucy Ndip.

**Supervision:** Sebastien Kenmoe, Lucy Ndip.

**Validation:** Martin Gael Oyono, Sebastien Kenmoe, Ngu Njei Abanda, Guy Roussel Takuissu, Jean Thierry Ebogo-Belobo, Raoul Kenfack-Momo, Cyprien Kengne-Nde, Donatien Serge Mbaga, Serges Tchatchouang, Josiane Kenfack-Zanguim, Robertine Lontuo Fogang, Elisabeth Zeuko'o Menkem, Juliette Laure Ndzie Ondigui, Ginette Irma Kame-Ngasse, Jeannette Nina Magoudjou-Pekam, Arnol Bowo-Ngandji, Seraphine Nkie Esemu, Lucy Ndip.

**Writing – original draft:** Martin Gael Oyono, Sebastien Kenmoe.

**Writing – review & editing:** Martin Gael Oyono, Sebastien Kenmoe, Ngu Njei Abanda, Guy Roussel Takuissu, Jean Thierry Ebogo-Belobo, Raoul Kenfack-Momo, Cyprien Kengne-Nde, Donatien Serge Mbaga, Serges Tchatchouang, Josiane Kenfack-Zanguim, Robertine Lontuo Fogang, Elisabeth Zeuko'o Menkem, Juliette Laure Ndzie Ondigui, Ginette Irma Kame-Ngasse, Jeannette Nina Magoudjou-Pekam, Arnol Bowo-Ngandji, Seraphine Nkie Esemu, Lucy Ndip.

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
