## [Decision Letter · Decision Letter 0]

24 May 2022

Dear Dr Kenmoe,

Thank you very much for submitting your manuscript "Epidemiology of yellow fever virus in humans, arthropods, and non-human primates in sub-Saharan Africa: a systematic review and meta-analysis." for consideration at PLOS Neglected Tropical Diseases. As with all papers reviewed by the journal, your manuscript was reviewed by members of the editorial board and by several independent reviewers. The reviewers appreciated the attention to an important topic. Based on the reviews, we are likely to accept this manuscript for publication, providing that you modify the manuscript according to the review recommendations. 

The review and meta-analyzed article were well received by the reviewers; however, some issues require more attention. Additional details are needed in the statistical methods section 

The discussion maybe improved by expanding the section that compared the meta-analyzes with recent reports and the section pointing to the limitations of this study

Fig-2) It would may help to link the maps of Fig-2 with the graphs of Fig-3

Fig-3) Fig-3 is mostly a table, the graph shown is redundant with the data.

Sincerely,

Ernesto T. A. Marques, M.D./Ph.D

Associate Editor

Scott Weaver

Deputy Editor

The review and meta-analyzed article were well received by the reviewers; however, some issues require more attention. Additional details are needed in the statistical methods section 

The discussion maybe improved by expanding the section that compared the meta-analyzes with recent reports and the section pointing to the limitations of this study

Fig-2) It would may help to link the maps of Fig-2 with the graphs of Fig-3

Fig-3) Fig-3 is mostly a table, the graph shown is redundant with the data.

Reviewer's Responses to Questions

**Key Review Criteria Required for Acceptance?**

**Methods**

-Are the objectives of the study clearly articulated with a clear testable hypothesis stated?

-Is the study design appropriate to address the stated objectives?

-Is the population clearly described and appropriate for the hypothesis being tested?

-Is the sample size sufficient to ensure adequate power to address the hypothesis being tested?

-Were correct statistical analysis used to support conclusions?

-Are there concerns about ethical or regulatory requirements being met?

Reviewer #1: The objectives of the study are clearly stated.

The systematic review and meta-analysis design adopted in this study is appropriate to address the study objectives.

The study area covered, study population was explicitly described.

Sample size was not necessary for this study, since this was a systematic review of published literature. 

The descriptive, and meta-analysis methods were appropriate to analyze the data. Sensitivity and publication analyses were appropriate to address heterogeneity and publication bias. 

there are no concerns about ethical requirements. The review reports on previously published data, ethical clearance was not required.

Reviewer #2: 1)- The objectives of the study were generally articulated

2)- In general the section Mthods needs further imporvements. For exemple, the authors should:

a) describe in depth the statistical method of the meta-analysis (ex. fixed effect or rendom effect estimation).

b) provide sensitvity analysis to show the robustness of the results.

c)- provide the criteria and summary table (or figure) showing the quality of the studies should be given in the manuscript. This help to assess the comparability of the studies before performing the meta-analysis.

Reviewer #3: The methodology followed sufficient analysis. Clearly describing the entire methodology.

**Results**

-Does the analysis presented match the analysis plan?

-Are the results clearly and completely presented?

-Are the figures (Tables, Images) of sufficient quality for clarity?

Reviewer #1: the analysis presented marched the analysis plan 

Risk of bias: was well conducted to clearly ascertain the quality of included studies. 

Baseline description: the summary and individual data of the included studies was clearly presented as stated in the analysis plan

Random effect model: clearly done to estimate the CFR of YFV and the prevalence of YFV among humans and the results presented in forest plot

Cochran Q test and the I2 test statistic: results clearly reported as described in the analysis plan

Sensitivity analysis: sub-group analysis was conducted as stated in the analysis plan

publication bias and analysis: Funnel plots and Egger's regression test are presented as stated in analysis plan

the results and figures are clearly and completely presented 

the figures are of sufficient quality, well labeled and clear

Reviewer #2: 1)- The results fairly matched with the plan analysis

2)- The results should be compared with current results observed in the litterature. 

3)- The quality of figures may be improved.

Reviewer #3: The analyzes followed the methodological planning and presented very interesting results.

I emphasize that these results are of paramount importance for public health worldwide.

**Conclusions**

-Are the conclusions supported by the data presented?

-Are the limitations of analysis clearly described?

-Do the authors discuss how these data can be helpful to advance our understanding of the topic under study?

-Is public health relevance addressed?

Reviewer #1: All the conclusions are supported by the data presented

Limitations of the analysis are clearly stated (L356-L359)

Detailed discussion of how the data advances our understanding of the topic is clearly elaborated. 

the public health relevance of the study is addressed

Reviewer #2: 1)- The conclusions were generally clear.

2)- However, beyond the classical limits (heterogeneity, publication bais), the authors may provide others specific limitations of the study

Reviewer #3: The conclusion is supported by the data presented conclusively. The limitations of the study are clearly presented. This is a very important study for global public health as it is the first review in the region.

**Editorial and Data Presentation Modifications?**

Reviewer #1: The data was clearly presented in this article and there are modifications am suggesting to the authors.

Reviewer #2: (No Response)

Reviewer #3: I suggest accepting the article without changes.

**Summary and General Comments**

Reviewer #1: The authors followed all the necessary steps laid out in the PRISMA guideline and the systematic review and meta-analysis was clearly conducted.

All results were systematically presented reporting pooled and heterogeneity results. 

Issues around publication bias were reported and highlighted as part of the limitations

Reviewer #2: (No Response)

Reviewer #3: (No Response)

PLOS authors have the option to publish the peer review history of their article (what does this mean?). If published, this will include your full peer review and any attached files.

Reviewer #1: No

Reviewer #2: No

Reviewer #3: Yes: Thamiris D'Almeida Balthazar

Figure Files:

Data Requirements:

Reproducibility:

References

---

## [Decision Letter · Decision Letter 1]

27 Jun 2022

Dear Dr Kenmoe,

We are pleased to inform you that your manuscript 'Epidemiology of yellow fever virus in humans, arthropods, and non-human primates in sub-Saharan Africa: a systematic review and meta-analysis.' has been provisionally accepted for publication in PLOS Neglected Tropical Diseases.

Best regards,

Ernesto T. A. Marques, M.D./Ph.D

Associate Editor

Scott Weaver

Deputy Editor

Reviewer's Responses to Questions

**Key Review Criteria Required for Acceptance?**

**Methods**

-Are the objectives of the study clearly articulated with a clear testable hypothesis stated?

-Is the study design appropriate to address the stated objectives?

-Is the population clearly described and appropriate for the hypothesis being tested?

-Is the sample size sufficient to ensure adequate power to address the hypothesis being tested?

-Were correct statistical analysis used to support conclusions?

-Are there concerns about ethical or regulatory requirements being met?

Reviewer #1: The objectives of the study are clearly stated.

The systematic review and meta-analysis design adopted in this study is appropriate to address the study objectives.

The study area covered, study population was explicitly described.

Sample size was not necessary for this study, since this was a systematic review of published literature.

The descriptive, and meta-analysis methods were appropriate to analyze the data. Sensitivity and publication analyses were appropriate to address heterogeneity and publication bias.

there are no concerns about ethical requirements. The review reports on previously published data, ethical clearance was not required.

Reviewer #2: I have no additional comments. The responses of the authors to me responses are acceptable

Reviewer #3: All considerations were accepted and reviewed, and I am satisfied with the article.

**Results**

-Does the analysis presented match the analysis plan?

-Are the results clearly and completely presented?

-Are the figures (Tables, Images) of sufficient quality for clarity?

Reviewer #1: the analysis presented marched the analysis plan

Risk of bias: was well conducted to clearly ascertain the quality of included studies.

Baseline description: the summary and individual data of the included studies was clearly presented as stated in the analysis plan

Random effect model: clearly done to estimate the CFR of YFV and the prevalence of YFV among humans and the results presented in forest plot

Cochran Q test and the I2 test statistic: results clearly reported as described in the analysis plan

Sensitivity analysis: sub-group analysis was conducted as stated in the analysis plan

publication bias and analysis: Funnel plots and Egger's regression test are presented as stated in analysis plan

the results and figures are clearly and completely presented

the figures are of sufficient quality, well labeled and clear

Reviewer #2: I have no additional comments. The responses of the authors to my comments are acceptable.

Reviewer #3: All considerations were accepted and reviewed, and I am satisfied with the article.

**Conclusions**

-Are the conclusions supported by the data presented?

-Are the limitations of analysis clearly described?

-Do the authors discuss how these data can be helpful to advance our understanding of the topic under study?

-Is public health relevance addressed?

Reviewer #1: All the conclusions are supported by the data presented

Limitations of the analysis are clearly stated

Detailed discussion of how the data advances our understanding of the topic is clearly elaborated.

the public health relevance of the study is addressed

Reviewer #2: I have no additional comments. The responses of the authors to my comments are acceptable.

Reviewer #3: All considerations were accepted and reviewed, and I am satisfied with the article.

**Editorial and Data Presentation Modifications?**

Reviewer #1: The data was clearly presented in this article

no modifications suggested as in previous submission

Reviewer #2: Accept

Reviewer #3: I recommend accepting the article.

**Summary and General Comments**

Reviewer #1: The authors followed all the necessary steps laid out in the PRISMA guideline

All results were systematically presented

Issues around publication bias were reported

Reviewer #2: I have no additional comments. The responses of the authors to my comments are acceptable.

Reviewer #3: (No Response)

PLOS authors have the option to publish the peer review history of their article (what does this mean?). If published, this will include your full peer review and any attached files.

Reviewer #1: No

Reviewer #2: No

Reviewer #3: No

---

## [Editor Report · Acceptance letter]

7 Jul 2022

Dear Dr Kenmoe,

We are delighted to inform you that your manuscript, "Epidemiology of yellow fever virus in humans, arthropods, and non-human primates in sub-Saharan Africa: a systematic review and meta-analysis.," has been formally accepted for publication in PLOS Neglected Tropical Diseases.

Best regards,

Shaden Kamhawi

co-Editor-in-Chief

Paul Brindley

co-Editor-in-Chief
